# Development of a Multispecies Double-Antigen Sandwich ELISA Using N and RBD Proteins to Detect Antibodies against SARS-CoV-2

**DOI:** 10.3390/ani13223487

**Published:** 2023-11-12

**Authors:** Maritza Cordero-Ortiz, Mónica Reséndiz-Sandoval, Freddy Dehesa-Canseco, Mario Solís-Hernández, Jahir Pérez-Sánchez, Carlos Martínez-Borges, Verónica Mata-Haro, Jesús Hernández

**Affiliations:** 1Laboratorio de Inmunología, Centro de Investigación en Alimentación y Desarrollo, A.C., Hermosillo 83304, Sonora, Mexico; maritzacoordero@gmail.com (M.C.-O.); mresendiz@ciad.mx (M.R.-S.); 2Comisión México-Estados Unidos para la Prevención de la Fiebre Aftosa y otras Enfermedades Exóticas de los Animales (CPA), Servicio Nacional de Sanidad, Inocuidad y Calidad Agroalimentaria (SENASICA), Secretaría de Agricultura y Desarrollo Rural (SADER), Ciudad de Mexico 05110, Mexico State, Mexico; freddy_1994333@hotmail.com (F.D.-C.); mario.solis@senasica.gob.mx (M.S.-H.); 3Centro de Biotecnología Genómica, Instituto Politécnico Nacional, Cd., Reynosa 88710, Tamaulipas, Mexico; jperezs2200@alumno.ipn.mx; 4Hospital Veterinario Borges, Hermosillo 83600, Sonora, Mexico; borgesmca@veterinariasborges.com; 5Laboratorio de Microbiología e Inmunología, Centro de Investigación en Alimentación y Desarrollo, A.C., Hermosillo 83304, Sonora, Mexico; vmata@ciad.mx

**Keywords:** SARS-CoV-2, COVID-19, ELISA, microneutralization, cats, multispecies

## Abstract

**Simple Summary:**

SARS-CoV-2 infects humans and several animals, but infection of companion and zoo animals occurs mainly from humans through the aerosol route. Constant monitoring of the infection in animals is essential because of their close contact with humans, which increases the risk of infection and the potential for a surge of new viral variants. Therefore, it is of relevance to develop economical, fast, and effective diagnostic tests that are able to detect previous infections in multiple animal species. This study aimed to produce a double-antigen sandwich enzyme-linked immunosorbent assay (ELISA) using N and RBD proteins. The results show that this assay is able to detect antibodies against SARS-CoV-2 in serum samples from cats, dogs, guinea pigs, rats, tigers, and humans.

**Abstract:**

SARS-CoV-2 infects humans and a broad spectrum of animal species, such as pets, zoo animals, and nondomestic animals. Monitoring infection in animals is important in terms of the risk of interspecies transmission and the emergence of new viral variants. Economical, fast, efficient, and sensitive diagnostic tests are needed to analyze animal infection. Double-antigen sandwich ELISA has the advantage of being multispecies and can be used for detecting infections caused by pathogens that infect several animal hosts. This study aimed to develop a double-antigen sandwich ELISA using two SARS-CoV-2 proteins, N and RBD. We compared its performance, when using these proteins separately, with an indirect ELISA and with a surrogate virus neutralization test. Positive and negative controls from a cat population (*n* = 31) were evaluated to compare all of the tests. After confirming that double-antigen sandwich ELISA with both RBD and N proteins had the best performance (AUC= 88%), the cutoff was adjusted using positive and negative samples from cats, humans (*n* = 32) and guinea pigs (*n* = 3). The use of samples from tigers (*n* = 2) and rats (*n* = 51) showed good agreement with the results previously obtained using the microneutralization test. Additionally, a cohort of samples from dogs with unknown infection status was evaluated. These results show that using two SARS-CoV-2 proteins in the double-antigen sandwich ELISA increases its performance and turns it into a valuable assay with which to monitor previous infection caused by SARS-CoV-2 in different animal species.

## 1. Introduction

SARS-CoV-2 belongs to the *Coronaviridae* family and the *Betacoronavirus* genus [1] and can infect several animal hosts, including domestic animals, such as cattle and small ruminants [2,3], companion animals such as cats and dogs [4,5,6], zoo animals such as tigers, lions and white rhinoceros [7,8,9], sewer rats [10,11] and wild animals such as deer [12,13]. Until now, evidence has indicated that SARS-CoV-2 transmission mainly occurs from humans to companion and zoo animals, which have the highest risk because of their close contact with owners and caretakers [14,15]. Monitoring exposure in animals that constantly interact with humans is essential in order to keep track of SARS-CoV-2 transmission between them. To achieve the correct surveillance of such infections, it is imperative to use reliable tests in the diagnosis of SARS-CoV-2 in different animals. The plaque reduction neutralization test (PRNT) is the gold standard for detecting the presence of neutralizing antibodies against SARS-CoV-2. In addition, the microneutralization test (MNT) is a very accurate alternative that allows the evaluation of more samples and is easier to perform [16]. However, these assays are time-consuming, requiring level 3 biosafety laboratory (BSL3) conditions and well-trained personnel.

Enzyme-linked immunosorbent assay (ELISA) is a fast and economical serological test that only requires level 2 biosafety laboratory (BSL2) conditions [17]. Indirect ELISA is one of the most commonly used assays with which to detect antibodies against SARS-CoV-2, and the nucleocapsid (N), spike (S), or receptor-binding domain (RBD) are the most common proteins used in this test. Double-antigen sandwich ELISA has also been widely used in monitoring other infectious diseases [18,19] but is not commonly used in SARS-CoV-2 infections. This assay can detect antibodies from different animals with a higher diagnostic sensitivity as it detects all antibody isotypes. These characteristics make double-antigen sandwich ELISA more helpful in the detection of zoonotic pathogen antibodies.

There are reports of the use of a commercial double-antigen sandwich ELISA test that uses the N protein. Barua et al. (2021) have found a good correlation between the commercial double-antigen sandwich ELISA test and the surrogate virus neutralization test (sVNT) with negative samples, but only 3 of 11 samples that were positive for the commercial double-antigen sandwich test also tested positive for the surrogate virus neutralization test. The lack of antibodies against the RBD and the possible cross-reaction to other coronaviruses may explain these results [20]. Similar results have been reported by Adler et al. (2022) when comparing the commercial double-antigen sandwich test with an indirect immunofluorescence test (iIFT) and sVNT. A good correlation was found between iIFT and sVNT, in contrast with the commercial double-antigen sandwich test using N protein [21]. Currently, there are limited options for double-antigen sandwich ELISA tests in terms of the detection of antibodies against SARS-CoV-2. For these reasons, we aimed to develop a double-antigen sandwich ELISA using N and RBD proteins. We compared its diagnostic sensitivity, specificity and area under the curve (AUC), based on the receiver operating characteristic (ROC) curve, with a double-antigen sandwich ELISA that used only RBD or N proteins; with an indirect ELISA using RBD, N or S1 proteins; and with a surrogate virus neutralization test (sVNT).

## 2. Materials and Methods

### 2.1. Serum Samples

Thirty-one serum samples from cats, thirty-two from humans, and three from guinea pigs were used as positive and negative controls to create ROC curves and determine the cutoff. Additionally, sera from two tigers, fifty-one rats, and forty-five dogs were evaluated. Cat and dog samples were remnants from clinical analyses of two veterinary clinics in Hermosillo, Sonora, Mexico. Samples from guinea pigs have been previously described [22]. One guinea pig was immunized with the ancestral strain, another with the subvariant Omicron BA.1, and a third remained as a negative control. Human serum samples included pre-pandemic samples collected before 2020 and samples from individuals who were only vaccinated, infected and vaccinated, or only infected. These samples have been described and evaluated in previous studies [23,24,25]. Tiger sera were collected in 2021 by the Mexico–United States Commission for the Prevention of Foot-and-Mouth Disease and Other Exotic Animal Diseases (CPA) of the National Service for Disease Control, Safety, and Agri-food Quality (SENASICA) in Mexico City as part of the passive surveillance of SARS-CoV-2 (Case number: CPA-06880-21). Rat sera were collected from sewer rats from a local market in the municipality of Reynosa and Matamoros in the state of Tamaulipas. All serum samples were stored at −20 °C until analysis and were processed in a level II biosafety hood. This study was evaluated and approved by the CIAD ethics committee (Approval number CEI/005-2/2020), the Bioethics Committee of the National School of Medicine and Homeopathy of the National Polytechnic Institute (Approval number CBE/006/2020) and the CPA-SENASICA ethics committee (Approval number CICUAL-CPA-001-2022).

### 2.2. Microneutralization Test

Cat, tiger, and rat serum samples were evaluated using the microneutralization test as previously described [22]. Samples were inactivated using gamma radiation at 56 °C for one hour. Then, twofold serial dilutions were made from 1:10 to 1:5120. Afterward, a 50 µL sample of the virus (100 TCID_50_) was added and incubated at 37 °C and with 5% CO_2_ for 60 min, and the serum–virus mix was transferred to 96-well plates with Vero cells. Finally, the results were read and interpreted.

### 2.3. Production of Recombinant Proteins

The genes used to produce the recombinant proteins were synthetized and inserted into a pcDNA3.1(+) plasmid by GenScript (GenScript, Piscataway, NJ, USA). The genes used in this study were the extracellular domain of hACE2, SARS-CoV-2 S1 protein, SARS-CoV-2 N protein, and SARS-CoV-2 RBD. All constructs were preceded by a signal peptide and a His-tag (6×His) C-terminal domain.

The recombinant proteins were produced using the Expi293 Expression System following the manufacturer’s instructions (Thermo Fisher Scientific, Waltham, MA, USA). Expi293 cells (75 × 106) were grown in 125 mL polycarbonate Erlenmeyer flasks (Corning, Corning, NY, USA) and Expi293 expression medium. Cells were transfected with 30 µg of plasmid and 81 µg of Expifectamine 293 reagent (both diluted in OptiMEM-I), incubated at 37 °C and 125 rpm with 8% CO_2_ and Expifectamine 293 Transfection Enhancers 1 and 2 were added after 20 h. After five days of culture, the supernatant was harvested and clarified by centrifugation at 4000× *g* for 10 min, filtered through a 0.22 µm filter, and proteins were purified in the chromatograph ÄKTA GO (GE Healthcare Life Sciences, Marlborough, MA, USA) using a linear gradient from 0 to 100% elution buffer (Imidazole 500 mM, NaCl 500 mM, and 20 mM NaH_2_PO_4_). Finally, the proteins were quantified, concentrated, desalted, aliquoted by dilution in PBS pH 7.4 and stored at −80 °C until use.

### 2.4. Double-Antigen ELISA

Ninety-six-well maximum adherence plates (Thermo Fisher Scientific) were coated with either N, RBD, or N and RBD together. These proteins were produced in our laboratory as described in Section 2.3. The N and RBD proteins were diluted in Antigen Coating Buffer 5X (ImmunoChemistry Technologies, Davis, CA, USA) at 3 µg/mL and 2 µg/mL, respectively. Additionally, both proteins (N and RBD) were coated together in each well at a concentration of 1 µg/mL and at a proportion of 1:1. To coat the plates, 50 µL of the diluted proteins were added to each well and incubated for 24 h at room temperature (25 °C, RT). Then, the wells were blocked with General Block (ImmunoChemistry Technologies) for 24 h at RT. Samples were diluted 1:1 with General Sample Diluent (ImmunoChemistry Technologies), and 50 µL of the dilution were incubated for 1 h at 37 °C. Subsequently, 50 µL of N and RBD proteins conjugated with peroxidase (N-HRP, RBD-HRP) were added to each plate as appropriate (i.e., plates coated with N protein were revealed with N-HRP) at the same concentration as the coated proteins and were incubated for 30 min at RT. Afterward, 50 µL of the 3,3′,5,5′-tetramethylbenzidine (TMB, 1-Component HRP Microwell Substrate, ImmunoChemistry Technologies) reagent was added and incubated for 20 min in darkness at RT. To stop the reaction, 50 µL of 1 M H_2_SO_4_ was added. Finally, plates were read at 450 nm (Thermo Scientific Multiskan FC, Waltham, MA, USA).

### 2.5. Indirect ELISA

Indirect ELISA was used to detect IgG antibodies against the N, RBD, or S1 proteins of SARS-CoV-2, previously produced in our laboratory as described in Section 2.3. Ninety-six-well maximum adherence plates (Thermo Fisher Scientific) were coated with recombinant N, S1, or RBD proteins, as appropriate. The coating was performed by diluting the protein with carbonate buffer (0.01 M NaHCO_3_ and 0.0875 M Na_2_CO_3,_ pH 9) to a concentration of 2 µg/mL and incubating at 4 °C overnight. Then, the plates were washed with 250 µL of phosphate buffer solution (PBS, 0.8% NaCl, 0.115% Na_2_HPO_4_, 0.02% KH_2_PO_4_, 0.02% KCl) and blocked with 100 µL of blocking buffer (2% bovine serum albumin, 0.02% sodium azide, and 3% glucose in PBS with 0.05% Tween 20) (Sigma Aldrich, St. Louis, MI, USA) for one hour at RT. Serum samples were diluted 1:150 with 1% milk (NFDM, Non-Fat Dry Milk Omniblok, AmericanBio, Canton, MA, USA), and 50 µL were added to each well and incubated for 30 min. Then, 50 µL of peroxidase-conjugated anti-cat IgG secondary antibody (goat anti-Cat HRP conjugate, BETHYL, Montgomery, TX, USA) diluted 1:50,000 with PBS were added and incubated for 30 min. Finally, 50 µL of TMB were added and incubated for 20 min, and the reaction was stopped with 50 µL of 1 M H_2_SO_4_. The plates were read at 450 nm (Thermo Scientific Multiskan FC).

All incubations were performed under constant agitation, darkness, and at RT. Five washes were performed between incubations with 250 µL of 0.1% Tween 20 in PBS. These conditions were maintained for indirect ELISA using any N, RBD, or S1 protein.

### 2.6. Surrogate Virus Neutralization Test

An in-house surrogate virus neutralization test (sVNT) was developed to evaluate the presence of neutralizing antibodies in cat samples. A 96-microtube plate (Corning) was used to dilute serum samples with PBS (1:10) in a final volume of 60 µL. Then, 60 µL of RBD-HRP (400 ng/mL) were added to the samples and incubated for 30 min at 37 °C. Then, 100 µL of the serum and the RBD–HRP mixture were incubated in a plate with recombinant ACE-2 for 15 min at 37 °C. The plate was then washed three times with 250 µL of 0.1% Tween 20 in PBS. Then, 100 µL of TMB were incubated for 20 min in darkness and at RT. To stop the reaction, 50 µL of H_2_SO_4_ were added. The reading was performed in a spectrophotometer (Thermo Scientific Multiskan FC) at a wavelength of 450 nm. To determine the neutralization percentage, the following formula was used:$$\%N=\left(1-\left(\frac{Sample\;OD}{Negative\;Control\;OD}\right)\right)\times 100$$
where %N stands for neutralization percentage, and OD stands for optical density.

### 2.7. Repeatability and Reproducibility

Repeatability and reproducibility were determined for the double-antigen sandwich ELISA using RBD and N proteins. Repeatability was evaluated using four samples: one high positive sample from a vaccinated and infected human, one low positive sample from another vaccinated and infected human, and two negative samples from cats. Eleven complete runs of the test were performed, each representing one repetition.

Reproducibility was evaluated in two different laboratories (Laboratory 1 and Laboratory 2), and each had a different operator for the test. Operators were not informed about the positive or negative status of each sample (blind study). The operators of reproducibility were different from the operators of repeatability. Twenty identical aliquots of serum samples from cats (*n* = 7) and humans (*n* = 13) represented five high-positive samples (absorbance between 2 and 3), four low-positive samples (absorbance from 0.4 to 0.9), five high-negative samples closer to the cutoff (absorbance from 0.1 to 0.2) and five low-negative samples (absorbance ≤ 0.1).

### 2.8. Statistical Analysis

The cutoffs for the indirect ELISA, the double-antigen ELISA and the sVNT were determined using cat samples, ROC curves, diagnostic sensitivity and specificity, and AUC. The cutoff and ROC curves of the double-antigen sandwich ELISAs with RBD and N proteins were adjusted by adding the values of human and guinea pig serum samples as positive and negative controls. These analyses were performed with a significance level of 0.05 in the statistical analysis package GraphPad PRISM version 8.0.2.

## 3. Results

### 3.1. Microneutralization Test

Thirty-one samples from cats were evaluated. To identify accurate positive controls for this study, serum samples were assessed using MNT, as several studies have confirmed that most cats infected with SARS-CoV-2 produce neutralizing antibodies [22,26,27,28]. The analysis confirmed 5 positive samples with titers of 1:10 (*n* = 3), 1:20 (*n* = 1) and 1:40 (*n* = 1) and 26 samples without neutralizing antibodies against SARS-CoV-2. These samples were used as accurate positive and negative controls.

### 3.2. Double-Antigen Sandwich ELISA

Double-antigen sandwich ELISA has been previously used to detect antibodies against several pathogens [18,19,29,30,31], including SARS-CoV-2 [20,21,32,33,34]. Most double-antigen sandwich ELISAs use one protein to capture and detect antibodies. In this study, we evaluated three different formats of double-antigen sandwich ELISA: 1, using RBD protein; 2, using N protein; and 3, using both RBD and N proteins.

To establish the optimum concentration of capture and detection antigen in the double-antigen sandwich ELISA, we tested several concentrations of N protein using positive and negative human sera samples. For coating, we evaluated several concentrations of antigen: 0.5, 1, 2, 3, 4 and 5 µg/mL; and different concentrations of soluble antigen: 0.050, 0.10, 0.15, 0.2, 0.4, 0.8 and 1.6 µg/mL of N-HRP. Figure 1a shows that the maximal difference between the absorbances in the positive and negative controls was obtained with the 3 µg/mL capture and the 1.6 µg/mL detection antigen, respectively. Then, we evaluated two additional concentrations of detection antigen, 2 µg/mL and 3 µg/mL, and used high positive and low positive controls to find the combination with the best discrimination between the low positives and negatives. The combination of 3 µg/mL capture and detection presented the best performance (Figure 1b) and was further used for the double-antigen sandwich ELISA using N protein. In the case of the double-antigen sandwich ELISA using RBD, we used 2 µg/mL for capture and detection, which can differentiate negative controls and sera from only infected and from vaccinated and infected individuals (Figure 1c). These conditions were maintained for the double-antigen sandwich ELISA using RBD protein in further experiments.

After testing several combinations of capture and detection antigens, we concluded that using a higher concentration of antigen (capture and detection) had a minimum chance to discriminate between negative and low positive samples. Therefore, keeping the optimal concentration that was obtained separately for N and for RBD could increase the background of the ELISA test when using both proteins. Based on the results shown in Figure 1a, with 1 µg/mL of coated antigen, the absorbance remained similar to the optimal value (1 µg/mL). Consequently, we tested a 2 µg/mL (1 µg/mL of N and 1 µg/mL of RBD) capture and a 2 µg/mL (1 µg/mL of N and 1 µg/mL of RBD) soluble antigen. Figure 1d shows the results of the evaluation of 24 human serum samples: negative (*n* = 8), vaccinated and infected (*n* = 8) and infected nonvaccinated (*n* = 8). The results confirm that these conditions can discriminate negative from vaccinated and infected and from infected non-vaccinated.

Once the optimal protein concentrations for each test were established, the double-antigen sandwich ELISA was evaluated with positive and negative cat serum samples using RBD protein only, N protein only, and both RBD and N proteins. The results are expressed as absorbances and the optimal cutoff was determined with an ROC curve (Table 1). This test performed better when using RBD and N proteins in the double-antigen sandwich ELISA than when using RBD or N proteins separately (Table 1). The cutoff for the double-antigen sandwich ELISA with RBD and N proteins was established at 0.3615, with a diagnostic sensitivity and specificity of 80% and 100%, respectively, and an AUC of 88%. On the other hand, when using only the RBD protein, the cutoff was established at 0.3080, and this test showed the same diagnostic sensitivity values (80.0%) but a decreased diagnostic specificity (88.0%). It also had a slightly lower AUC value (86.4%). For the double-antigen sandwich ELISA using N protein, the cutoff was set at 0.2665, and the diagnostic sensitivity and specificity were 60% and 85.19%, respectively, while the AUC was 67.41%. These results show that, under our experimental conditions, the lowest performance was obtained using the N protein.

### 3.3. Indirect ELISA

Indirect ELISA has been used to detect antibodies against SARS-CoV-2 and other pathogens in several reports [35,36,37,38,39,40]. This study evaluated three proteins with which to detect antibodies against SARS-CoV-2: the RBD, N, and S1 proteins. The results were expressed as absorbance, and the optimal cutoff was determined with the ROC curve (Table 2).

Indirect ELISA using the RBD protein performed better than indirect ELISA using the S1 or N protein. The cutoff for indirect ELISA with RBD was set at 0.7960, with the highest diagnostic sensitivity (80%) and AUC (85.7%) among indirect ELISA and a diagnostic specificity of 89.3%. On the other hand, when using the S1 protein, the cutoff was established at 1.3333 with a lower diagnostic sensitivity (60.0%) and a higher diagnostic specificity (100.0%) but a diminished AUC (75.4%). Finally, when using the N protein, the cutoff was set at 0.4510, and this test had the lowest diagnostic sensitivity (60%), specificity (85.19%), and AUC (67.41%). These results show inferior performance when using the N protein, as in the double-antigen sandwich ELISA.

### 3.4. Surrogate Virus Neutralization Test

In addition to ELISA tests, an sVNT was developed with which to evaluate the presence of neutralizing antibodies against SARS-CoV-2. sVNT has been used to detect neutralizing antibodies against SARS-CoV-2 in humans, cats, and other animal species [20,26,41]. In this study, we developed and compared an sVNT using RBD-HRP and ACE2 to detect potential neutralizing antibodies against SARS-CoV-2. The results were expressed as percentages, and the ROC curve was used to determine the optimal cutoff (Table 3). The cutoff established for sVNT was 16.9%. The diagnostic sensitivity and specificity were 80.0% (95% CI: 37.5 to 98.9) and 96.15% (95% CI: 81.1 to 99.8), respectively, and the AUC was 82.31% (95% CI: 53.8 to 100), *p* = 0.0241. Samples with the highest titers of 1:40 and 1:20 in MNT were positive for sVNT, as well as two samples with titers of 1:10. These data demonstrate a good performance of this test, though one that is not better than double-antigen sandwich ELISA with RBD and N or only RBD.

### 3.5. Double-Antigen Sandwich ELISA with RBD and N Proteins in Different Animal Species

Human and guinea pig samples were evaluated using the double-antigen sandwich ELISA with RBD and N proteins to corroborate its multispecies capacity since this test had the best diagnostic sensitivity (80%), specificity (100%), and AUC (88%). As described previously, serum samples from vaccinated (*n* = 8), infected (*n* = 8), or both vaccinated and infected (*n* = 8) humans were used as positive controls, while pre-pandemic human samples (*n* = 8) were used as negative controls. In addition, positive and negative sera from immunized (*n* = 2) and nonimmunized (*n* = 1) guinea pigs were also evaluated. With these results, the cutoff that was obtained using cat samples was adjusted while considering the effects of human and guinea pig true positive and negative controls and a new ROC curve was created. The results show that the cutoff remained at the same value that was obtained with cat samples (0.3615) but with a higher diagnostic sensitivity (83.87%) and a higher AUC (92.55%). The diagnostic specificity remained at 100% (Table 4). Using cat, human, and guinea pig samples improved the performance of the double-antigen sandwich ELISA that uses RBD and N proteins to capture and detect antibodies against SARS-CoV-2.

The double-antigen sandwich ELISA using RBD and N proteins was also used to evaluate samples from tigers (*n* = 2), rats (*n* = 51), and dogs (*n* = 45) (Figure 2). The MNT test has been previously used to evaluate neutralizing antibodies in tiger and rat samples. The results show that tiger samples were negative when using double-antigen sandwich ELISA, which coincided with the results obtained by MNT. All of the positive rat samples for MNT (*n* = 9) were also shown to be positive by ELISA, except for one that remained close to the cutoff. Negative rat samples (*n* = 42) for MNT were also shown to be negative by ELISA, except for three samples with values near the cutoff line. Finally, dog sera (not previously evaluated by MNT) showed 3 out of 45 dog samples with positive values using double-antigen sandwich ELISA. These results confirm that the double-antigen sandwich ELISA can detect antibodies in tiger, rat, and dog samples.

### 3.6. Repeatability and Reproducibility of the Double-Antigen Sandwich ELISA with RBD and N Proteins

Repeatability was evaluated during eleven runs of the test on different days with the same operator using four samples that represented high positive, low positive, high negative and low negative values. The coefficient of variation remained below 20% for the high positive, high negative and low negative samples but was 26.12% for the low negative and low positive samples (Table 5). All high and low positive samples remained positive in all of the the runs, and high and low negative samples remained negative in all of the runs. The WOAH terrestrial manual recommends that low positive samples and negative samples are allowed to have a CV above 15% [42].

Reproducibility was evaluated in two different laboratory conditions by two different operators. The results show a correlation coefficient of 0.8725 (*p* < 0.0001). All negative samples remained negative for both laboratories, and all high positive samples were positive for both laboratories. Three low positive samples were negative for Laboratory 1, and one of these was also negative for Laboratory 2 (Figure 3).

## 4. Discussion

Indirect ELISA is probably the most commonly used assay with which to evaluate anti-SARS-CoV-2 antibodies in pets [35,36,37,43,44,45]. However, the double-antigen sandwich ELISA is the most appropriate when the aim is to assess antibodies in several species, regardless of the antibody isotype. Usually, double-antigen sandwich ELISA uses a protein to capture and detect antibodies, increasing the diagnostic sensitivity and reducing false positives. In the case of SARS-CoV-2, fewer studies have used a double-antigen sandwich ELISA [20,32,33,34]. In this study, we developed and evaluated a double-antigen sandwich ELISA that uses two proteins, N and RBD, as capture and detection antigens, respectively. This assay detected anti-SARS-CoV-2 antibodies in samples from cats, dogs, guinea pigs, rats, tigers, and humans.

Diagnostic tests such as RT–PCR can determine an active infection in different animal species and are important to determine acute infections [46,47]. However, the period to collect samples from infected animals and detect viral RNA in this test is usually short [48,49]. In contrast, serological assays, such as ELISA tests, allow the determination of previous infections, and as antibodies usually last longer in the animal bloodstream [50,51], these tests are useful for the surveillance of SARS-CoV-2 infection and epidemiological studies. PRNT and MNT have been considered the gold standard assays with which to evaluate antibodies against SARS-CoV-2. With this knowledge, and considering the high susceptibility of felines to SARS-CoV-2 and their ability to produce a robust neutralizing antibody response after natural or experimental infection with SARS-CoV-2 [28,49], we evaluated 31 cat samples with MNT to identify true-positive and true-negative controls. Five of the thirty-one samples were positive for MNT and were used to define cutoffs with ROC curves for all of the double-antigen and indirect ELISAs developed. A commercial double-antigen sandwich ELISA uses N protein to capture and detect antibodies. In this study, we evaluated three different formats and found that the combination of N and RBD proteins provided the highest performance, contrary to the use of N or RBD alone. We hypothesized that the combined N and RBD proteins could detect a higher range of antibodies, representing an advantage because the dynamics of the antibody response against these proteins seem different [52].

Optimization of double-antigen sandwich ELISA needed evaluation of several concentrations of capture and detection antigen. It was interesting to observe that lower concentrations of capture antigen with low detection antigen showed better behavior. However, higher concentrations of capture and detection provide a better overall performance. This suggests that the test requires an equilibrium between the capture and detection antigens, and in this way, we set the test using a 3 mg/mL capture and a 3 mg/mL detection antigen. Additionally, different human samples were evaluated to increase the power of the double-antigen sandwich ELISA with RBD and N proteins: negative, infected without vaccine, vaccinated without infection, and both vaccinated and previously infected. In addition, sera from negative and experimentally immunized guinea pigs were evaluated, and the results were used along with those from the cats to set the best cutoff based on the ROC curve. Guinea pigs have been commonly used as an animal model with which to evaluate novel vaccines against SARS-CoV-2 and have shown seroconversion through immunization with this virus [53,54]. The performance of our assay using these three types of samples showed the same cutoff set as cats (0.3615) but with an increased diagnostic sensitivity of 83.87% (CI 95%: 67.4 to 92.2), a diagnostic specificity of 100% (CI 95%: 89.8 to 100) and a higher AUC of 92.5 (CI 95%: 84.5 to 99.9) (*p* ≤ 0.0001).

An indirect ELISA using RBD, S1, or N proteins as antigens was also evaluated. The indirect ELISA using RBD showed the best results; however, though RBD is part of subunit 1 (S1) of the S protein, the performance of the test using this protein was lower, and its diagnostic sensitivity was very low (60%), indicating that not all of the MNT-positive samples were recognized as such. Thus, the true-positive cat samples used in this study mainly identified the RBD protein compared with the S1 domain. On the other hand, the use of N protein had the lowest performance (AUC close to 50%), contrary to other studies that show a tendency toward a higher diagnostic sensitivity in indirect ELISA using N protein rather than RBD in cats [35]. Other reports have established a good correlation between indirect ELISA using RBD and MNT or PRNT [37,44], as was observed in our study. Bold et al. (2022) have reported a similar diagnostic sensitivity and specificity for indirect ELISA using RBD or N proteins [38]. However, experimentally infected cats were evaluated in that report and those results differ from our results, in which a naturally infected population was evaluated. The use of the N protein to detect antibodies by an indirect ELISA seems difficult, in part due to the similarity of the SARS-CoV-2 N protein with canine coronaviruses that infect dogs [55]. To solve this problem, some authors have used immunoassays based on paramagnetic beads, such as xMAP (Luminex Corp), with diagnostic specificities of 96.5% for dogs and 100% for cats [56].

The sVNT evaluated in this study had good diagnostic sensitivity (80%) and specificity (96.2%) but was not higher than the double-antigen sandwich ELISA with RBD and N proteins. Good agreement between sVNT and PRNT has been reported in cat samples, with better diagnostic specificity than diagnostic sensitivity, similar to previous findings [57]. Currently, and to the best of our knowledge, there are reports of only two commercial double-antigen sandwich ELISAs: one that employs the N protein used in multiple species [20,32,33,34] and another that employs the RBD protein used in humans [57]. Low agreement between the results of commercial double-antigen ELISA with N protein and sVNT has been reported in cats [21] and dogs, using MNT as a reference test [20,33]. This is similar to our findings in ELISAs with only N protein, which had the lowest performance based on an MNT reference test. There are some controversial results with the use of N protein to detect antibodies in cats and dogs. In most cases, positive results come from studies evaluating experimentally infected animals, contrary to studies evaluating seroprevalence in animals with an unknown infection status [20,21,28]. An explanation of this phenomenon is that, after an experimental infection, animals produce an anti-N protein response that is easily detected. However, in natural infection, the anti-N response is low and of short duration. There are reports showing that some individuals can develop a low response to anti-N protein, especially in asymptomatic infections [58,59], and a similar scenario could occur in animals. Additionally, the cross-reactions with the N protein of coronavirus infecting cats and dogs could explain the discrepancies [55]. This could also explain why the use of RBD has better performance than N or even than S1.

Double-antigen sandwich ELISA using RBD and N proteins was also used to evaluate sera from rats, tigers, dogs, and animals susceptible to natural SARS-CoV-2 infection. All negative samples from rats and tigers with MNT were also negative with our test, except for three rat samples that remained slightly above the cutoff (0.363, 0.366, 0.374). All positive rat samples were also positive for ELISA except for one, with a value close to the cutoff (0.328). In the case of dog samples, we identified positive and negative results, suggesting that the assay can also be used in this species.

One of the limitations of this study is the small sample size, as it is ideal to have a large sample size that includes different species with which to evaluate the presence of antibodies and avoid bias in the results. However, with the samples available in this study, we can demonstrate the performance and comparison of these assays with multiple species that are susceptible to SARS-CoV-2, and the total number of controls (*n* = 66) is sufficient to have a primary validation of this test, according to the WOAH Terrestrial Manual, as it is difficult to obtain many samples from different species [42]. A second limitation of this study is that the infection status of cats and dogs was unknown, and that these samples were remnants from clinical analyses. It was not possible to evaluate dog samples by MNT and define them as positive or negative controls. However, according to our results and the performance of the double-antigen sandwich ELISA method, we can state that these samples are not false positives.

## 5. Conclusions

Our results show excellent discrimination in the evaluated samples and prove that this test can detect antibodies against SARS-CoV-2 in multiple species. It would have been ideal to have a more significant number of samples from other animals to corroborate these results, the lack of which represents one limitation of our study. However, we can state that the double-antigen sandwich ELISA with RBD and N proteins developed in this study is an excellent assay with which to seek antibodies against SARS-CoV-2 in several species. The assay is sensitive, specific, and valuable in cats, humans, tigers, guinea pigs, rats, and potentially other species. This assay can be used to perform surveillance studies in several animal species, including captive, domestic, and wild animals, in order to understand SARS-CoV-2 dissemination in animals and understand the role of these species as reservoirs of the virus.

## Figures and Tables

**Figure 1 animals-13-03487-f001:**
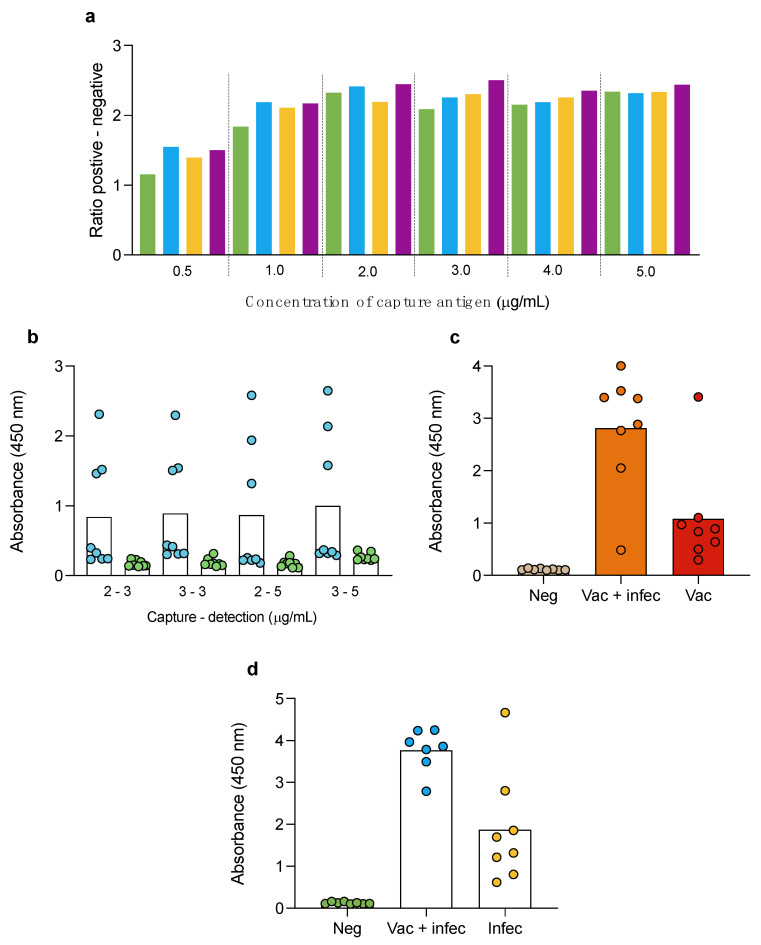
Optimization of the double-antigen sandwich ELISA. (**a**) Several concentrations of capture antigen (0.5 to 5.0 µg/mL) and detection antigen (0.2 (green bars), 0.4 (light blue bars), 0.8 (yellow bars), and 1.6 (purple bars)) were used to find the optimal combination of capture and detection antigen (N protein). Data are visualized to represent the maximal absorbance resulting from subtracting the mean of the positive controls’ absorbance minus the negative controls’ mean absorbance. In this experiment, human serum samples were evaluated. (**b**) To find the optimal conditions of the double-antigen sandwich ELISA using N protein, additional concentrations of capture (2 and 3 µg/mL) and detection (3 and 5 µg/mL) antigen were evaluated. Blue dots represent the positive controls, and the green dots represent the negative controls. (**c**) Optimization of the double-antigen sandwich ELISA using RBD with 2 µg/mL capture and detection antigen and sera from negative controls (Neg, light brown dots), individuals vaccinated and infected (Vac + infect, orange bar and dots), and vaccinated individuals red bar and dots. (**d**) Optimization of the double-antigen sandwich ELISA using N and RBD as capture (2 µg/mL, 1 µg/mL of N, and 1 µg/mL of RBD) and detection (2 µg/mL, 1 µg/mL of N and 1 µg/mL of RBD) antigens. In this case, we used sera from negative (Neg, green dots and bars), vaccinated and infected individuals (Vac + infect, light blue dots and bars), and infected (Infect, pale yellow dots and bars) humans.

**Figure 2 animals-13-03487-f002:**
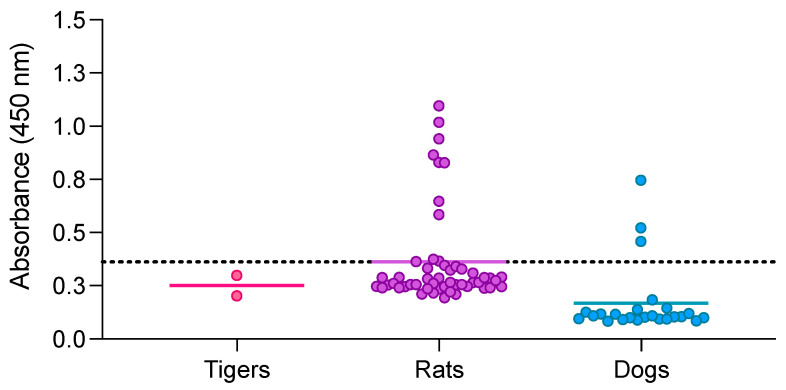
Double-antigen sandwich ELISA using RBD and N proteins was used to evaluate samples from tigers, rats, and dogs. Serum samples from tigers (pink dots), rats (purple dots), and dogs (blue dots) were assessed. Each point represents a sample from tigers (*n* = 2), rats (*n* = 51), or dogs (*n* = 45) whereas each line within groups represent the mean. The dotted line indicates the cutoff established at 0.3615.

**Figure 3 animals-13-03487-f003:**
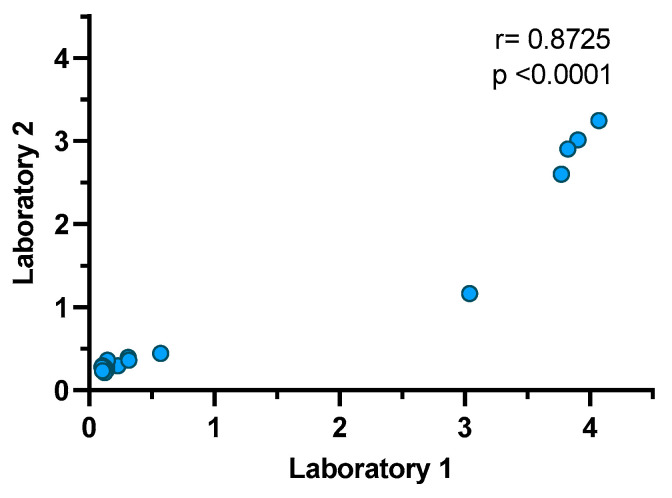
Correlation coefficient of reproducibility analysis. Laboratory 1 and Laboratory 2 evaluated 20 identical samples, including positive (*n* = 9) and negative (*n* = 11) samples, using the double-antigen sandwich ELISA with RBD and N proteins.

**Table 1 animals-13-03487-t001:** Diagnostic sensitivity and specificity of double-antigen sandwich ELISA.

	Cutoff	Diagnostic Sensitivity	Diagnostic Specificity	AUC	*p* Value
Format 1 (RBD protein)	0.3080	80% (CI 95%: 35.5 to 98.9)	88% (CI 95%: 70.0 to 95.8)	86.40% (CI 95%: 66.2 to 100)	0.0113
Format 2 (N protein)	0.2665	60% (CI 95%: 23.0 to 92.9)	85.19% (CI 95%: 67.5 to 94.1)	67.41% (CI 95%: 38.7 to 96)	0.2226
Format 3 (RBD and N protein)	0.3615	80% (CI 95%: 37.5 to 98.9)	100% (CI 95%: 86.7 to 100)	88% (CI 95%: 66.3 to 100)	0.0082

**Table 2 animals-13-03487-t002:** Diagnostic sensitivity and specificity of indirect ELISA.

	Cutoff	Diagnostic Sensitivity	Diagnostic Specificity	AUC	*p* Value
iELISA with RBD protein	0.7960	80% (CI 95%: 37.5 to 98.9)	89.29% (CI 95%: 72.8 to 96.3)	85.71% (CI 95%: 64.1 to 100)	0.0121
iELISA with S1 protein	1.3333	60% (CI 95%: 23.1 to 92.9)	100% (CI 95%: 87.1 to 100)	75.38% (CI 95%: 47.1 to 100)	0.0763
iELISA with N protein	0.4510	60% (CI 95%: 23.1 to 92.9)	69.23% (CI 95%: 50.0 to 83.5)	53.46% (CI 95%: 23.6 to 83.3)	0.8090

**Table 3 animals-13-03487-t003:** Diagnostic sensitivity and specificity of a surrogate virus neutralization assay.

	Cutoff	Diagnostic Sensitivity	Diagnostic Specificity	AUC	*p* Value
sVNT	16.9%	80.0 (CI 95%: 37.5 to 98.9)	96.15 (CI 95%: 81.1 to 99.8)	82.31 (CI 95%: 53.8 to 100)	0.0241

**Table 4 animals-13-03487-t004:** The diagnostic sensitivity and specificity of double-antigen sandwich ELISA using RBD and N proteins based on cats, humans, and guinea pigs.

Cutoff	Diagnostic Sensitivity	Diagnostic Specificity	AUC	*p* Value
0.3615	83.87 (CI 95%: 67.4 to 92.9)	100 (CI 95%: 89.8 to 100)	92.55 (CI 95%: 85.2 to 99.9)	<0.0001

**Table 5 animals-13-03487-t005:** The coefficient of variation of double-antigen sandwich ELISA using RBD and N proteins in the repeatability test.

Sample	Absorbance Mean	%CV
High positive	4.163	10.64
Low positive	0.6043	26.12
High negative	0.1781	12.53
Low negative	0.1488	17.37

## Data Availability

Data are contained within the article.

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
