# Peer review of "Development of a Multispecies Double-Antigen Sandwich ELISA Using N and RBD Proteins to Detect Antibodies against SARS-CoV-2"

_animals, 2023, doi:10.3390/ani13223487_

Round 1

Reviewer 1 Report (Previous Reviewer 3)

Comments and Suggestions for Authors The authors have adequately addressed the questions and comments from my initial review and I recommend that article be published.

Author Response

Thank you for your comments.

Reviewer 2 Report (Previous Reviewer 2)

Comments and Suggestions for Authors

The authors revised and improved the manuscrpt. However, I would be grateful to the authors if they could highlight the answers to my question in the manuscript. Specifically,  the letter they stated:

Answer: Regarding the size of positive samples, we have included a discussion to this aspect as suggested.

Answer: We agree the Reviewer comment, the potential of our assay have been underlined as suggested

Please highlight these answers, I didn’t find these changes in the text.

Author Response

Reviewer 1

The authors revised and improved the manuscript.

Answer: Thank you for your comments.

However, I would be grateful to the authors if they could highlight the answers to my question in the manuscript. Specifically, the letter they stated:

Regarding the size of positive samples, we have included a discussion to this aspect as suggested.

Answer: We have highlighted the first point in the discussion section, lines 490-496

We agree the Reviewer comment, the potential of our assay have been underlined as suggested.

Answer: We have highlighted the first point in the discussion section, lines 408-414

Please highlight these answers, I didn’t find these changes in the text.

Answer: Done as suggested.

Reviewer 3 Report (New Reviewer)

Comments and Suggestions for Authors

The work entitled "Development of a multispecies double-antigen sandwich ELISA using N and RBD proteins to detect antibodies against SARS-CoV-2 " and written by Cortero-Ortiz et al., describes the setup of a new multispecies ELISA kit that uses two different antigens to identify anti-SARS-CoV-2 antibodies in animals and humans. The work is of great interest, and the experimental design seems correct and stable from the point of view of scientific rigor. The work is potentially publishable. However, especially in describing the results (and in the figures), the authors have generated confusion in the reader. Below are my specific comments that require major revision.

Simple summary: The authors wrote: ”…but in companion and zoo animals…”. Does this mean that in other species, the infection is not due to contact with infected humans? Please specify it.

Abstract:

Line 31: The authors wrote, ”…ideal in zoonotic infections..”. In reality, this type of test is useful for infections that have a broad host spectrum (for example, SBV infection is not a zoonosis but can affect numerous species, and therefore multispecies commercial tests exist). Edit please.

Line 33 and 37-38: Here begins the confusion that the authors generate in the reader. Here it can be seen that the positive and negative samples were sampled from cats. Then the performances were evaluated for other species. My advice is to list all the positive and negative samples based on the species (and specify how they were determined as such) and think in a "multispecies" way (evaluating the performances of the total samples).

Line 48: The list of susceptible animals in which antibodies have been found with multispecies kits also includes cattle (DOI: 10.3390/ani12111459), small ruminants (DOI: 10.1007/s11259-022-10044-3) (among domestic animals), white rhinoceros (doi.org/10.3390/ani13162593 ) (among zoo animals), and several wild animals (as deer). Please modify it.

Some parts of the manuscript are in red; please correct them.

Matherials and methods:

Lines 90-91: The authors wrote: “Thirty-one serum samples from cats, 32 from humans, and 3 from guinea pigs were used as positive and negative controls to create ROC curves”. Only this? And the others?

Lines 97-98: How were these things evaluated? Medical history, laboratory tests, or both?

Line 104: How are these sera defined as positive and negative? Which reference test was used?

Line 121: The authors should clarify how they produced the antigens, at least in a preliminary manner; they cannot limit themselves to just providing the bibliographic reference. Also, because in a subsection (Production of Recombinant Proteins), you wrote about plasmids, etc. and this generates confusion for the reader.

Line 125-131: I understand that a sort of criss-cross reaction to determine the quantities of antigen, secondary, etc. was carried out (Figures). Why isn't it written about here? Yet it is important to understand how the authors identified the quantities of antigen, serum dilution, and secondary

Line 139: As 121

Results:

Figure 1 is not necessary.

The (first) figure 2 (Optimization of the double-antigen sandwich ELISA) lacks a legend, and it is not clear whether these results were obtained in humans or in which species.

The second figure 2 (Double-antigen sandwich ELISA using RBD) does not make much sense due to the few samples analyzed (8 species for tigers). My advice is to carry out a single analysis for all species, a single figure, and a single table.

Discussion:

Compare Elisa performances with others but also with other methods, such as chemiluminescence methods that appear more sensitive and specific (doi: 10.3201/eid2707.203314).

Comments on the Quality of English Language

English is ok and clear

Author Response

Reviewer 3

The work entitled "Development of a multispecies double-antigen sandwich ELISA using N and RBD proteins to detect antibodies against SARS-CoV-2 " and written by Cortero-Ortiz et al., describes the setup of a new multispecies ELISA kit that uses two different antigens to identify anti-SARS-CoV-2 antibodies in animals and humans. The work is of great interest, and the experimental design seems correct and stable from the point of view of scientific rigor. The work is potentially publishable. However, especially in describing the results (and in the figures), the authors have generated confusion in the reader. Below are my specific comments that require major revision.

Simple summary: The authors wrote: ”…but in companion and zoo animals…”. Does this mean that in other species, the infection is not due to contact with infected humans? Please specify it.

Answer: The changes were done as suggested

Abstract:

Line 31: The authors wrote, ”…ideal in zoonotic infections..”. In reality, this type of test is useful for infections that have a broad host spectrum (for example, SBV infection is not a zoonosis but can affect numerous species, and therefore multispecies commercial tests exist). Edit please.

Answer: The changes were done as suggested

Line 33 and 37-38: Here begins the confusion that the authors generate in the reader. Here it can be seen that the positive and negative samples were sampled from cats. Then the performances were evaluated for other species. My advice is to list all the positive and negative samples based on the species (and specify how they were determined as such) and think in a "multispecies" way (evaluating the performances of the total samples).

Answer: The changes were done as suggested

Line 48: The list of susceptible animals in which antibodies have been found with multispecies kits also includes cattle (DOI: 10.3390/ani12111459), small ruminants (DOI: 10.1007/s11259-022-10044-3) (among domestic animals), white rhinoceros (doi.org/10.3390/ani13162593 ) (among zoo animals), and several wild animals (as deer). Please modify it.

Answer: We have added this information as suggested.

Some parts of the manuscript are in red; please correct them.

Answer: The text in red represent the changes to our manuscript suggested in the first round of revisions. In this new version, the text in red show the changes suggested in the second round or revision.

Materials and methods:

Lines 90-91: The authors wrote: “Thirty-one serum samples from cats, 32 from humans, and 3 from guinea pigs were used as positive and negative controls to create ROC curves”. Only this? And the others?

Answer: Dog samples were remnants from clinical analyses of a veterinary clinic, and it was not possible to analyze them by microneutralization test, in consequence, can’t be used to create ROC curves. In the case of the tigers’ samples, only two negative samples were available. Rat samples were previously analyzed by microneutralization, with a positive and negative status. In this case, we decided to use these samples to confirm that the cutoff set with the multispecies samples can discriminate rat samples.

Lines 97-98: How were these things evaluated? Medical history, laboratory tests, or both?

Answer: Briefly, prepandemic samples were collected from a previous breast cancer project between the years 2017 to 2019 (DOI: 10.1111/tbed.14344). Samples from only vaccinated as well as infected and vaccinated individuals were obtained from adults vaccinated with a single dose of Ad5-nCoV (n=5) and a booster eight months later with the mRNA-1273 vaccine (n=3). The infection status was determined based on a demographic survey applied to participants, including previously infected individuals vaccinated with a single dose of Ad5-nCoV (n=4) and with a booster of mRNA-1273 (n=4) (DOI: 10.1038/s41392-023-01447-y). Samples from only infected individuals were obtained from patients with a PCR-confirmed SARS-CoV-2 infection, 3 to 8 weeks after the onset of symptoms (https://doi.org/10.1016/j.isci.2023.106562). 

The changes in the manuscript were done as follow: Human serum samples included prepandemic samples collected before 2020 and samples from only vaccinated, infected and vaccinated, and only infected individuals. These samples were described and evaluated in previous studies done by our group [23-25].

Line 104: How are these sera defined as positive and negative? Which reference test was used?

Answer: Microneutralization test was used to evaluate tigers’ and rats’ samples. PMID: 37375525

Line 121: The authors should clarify how they produced the antigens, at least in a preliminary manner; they cannot limit themselves to just providing the bibliographic reference. Also, because in a subsection (Production of Recombinant Proteins), you wrote about plasmids, etc. and this generates confusion for the reader.

Answer: The subsection “Production of Recombinant Proteins” was modified accordingly.

Line 125-131: I understand that a sort of criss-cross reaction to determine the quantities of antigen, secondary, etc. was carried out (Figures). Why isn’t it written about here? Yet it is important to understand how the authors identified the quantities of antigen, serum dilution, and secondary

Answer: Lines 237-262 describe the antigen concentrations used in this study. In the case of indirect ELISA, our group has previously reported the conditions for this test. For the Surrogate Virus Neutralization Test, we followed the conditions previously reported with some modifications.

Line 139: As 121

Answer: The changes were done as suggested

Results:

Figure 1 is not necessary.

Answer: Figure 1 has been removed as suggested

The (first) figure 2 (Optimization of the double-antigen sandwich ELISA) lacks a legend, and it is not clear whether these results were obtained in humans or in which species.

Answer: The changes were done as suggested

The second figure 2 (Double-antigen sandwich ELISA using RBD) does not make much sense due to the few samples analyzed (8 species for tigers). My advice is to carry out a single analysis for all species, a single figure, and a single table.

Answer: We have modified figure 2 as suggested

Discussion:

Compare Elisa performances with others but also with other methods, such as chemiluminescence methods that appear more sensitive and specific (doi: 10.3201/eid2707.203314).

Answer: The reference has been added to the discussion section. Lines 457-461 in discussion section.

English is ok and clear

Thank you.

Reviewer 4 Report (New Reviewer)

Comments and Suggestions for Authors

Summary:

This study developed a double-antigen sandwich ELISA using N and RBD proteins to detect SARS-CoV-2 antibodies in various species including cats, dogs, guinea pigs, rats, tigers, and humans. Its performance was compared with other methods. Evaluations using positive and negative controls helped refine its accuracy. The results indicated that the double-antigen sandwich ELISA, incorporating two SARS-CoV-2 proteins, enhanced its efficacy and is useful for monitoring infections across different animal species. While the paper is fundamentally robust, there remain aspects that require further elucidation and attention.

Major comments:

1.    Please furnish the immunological records of the animals used in this study, to the extent of the author's knowledge. While lines 94-96 detail the immune records of a guinea pig, there is no mention of other animals, particularly domesticated ones such as cats and dogs.

2.    It has come to attention that two distinct figures are both labeled as "Figure 2", one on page 7 and the other on page 10. This has led to confusion among the readers. Please rectify this oversight.

3.    Pertaining to "Figure 2" on page 7, the concentration of the capture antigen concentration group at 0.5 ug/mL shows the bar at 0.4 (blue bar) being higher than at 0.8 (yellow bar). A similar observation is made for the 1.0 ug/mL and 2.0 ug/mL groups. Please provide an explanation and delve into a discussion on this phenomenon.

4.    In Table 1, the diagnostic sensitivity of the N protein stands at a mere 60%, with a p-value of 0.2226. It would be beneficial if the authors could discuss this in the subsequent section, perhaps shedding light on the reasons for the N protein's performance. Similarly, Table 2 indicates that the indirect ELISA with the S1 protein exhibits low diagnostic sensitivity. An explanation for this observation would be appreciated.

5.    Between lines 357 and 359, it is mentioned that 3 out of the 45 dog samples tested positive using the sandwich ELISA method. It would be informative to know whether these 3 samples had previously tested positive through other techniques, whether the respective dogs exhibited any positive symptoms, or if there's a possibility that these were false positives.

Minor Comments:

1.    "Microneutralization test" is abbreviated as both "MNT" and "MN test". It's recommended to consistently use one abbreviation throughout.

2.    Line 185: Insert a space before starting a new paragraph.

3.    Line 457: "ELISAS" should be corrected to "ELISAs".

Comments on the Quality of English Language

Moderate editing of English language required

Author Response

Reviewer 4

Summary:

This study developed a double-antigen sandwich ELISA using N and RBD proteins to detect SARS-CoV-2 antibodies in various species including cats, dogs, guinea pigs, rats, tigers, and humans. Its performance was compared with other methods. Evaluations using positive and negative controls helped refine its accuracy. The results indicated that the double-antigen sandwich ELISA, incorporating two SARS-CoV-2 proteins, enhanced its efficacy and is useful for monitoring infections across different animal species. While the paper is fundamentally robust, there remain aspects that require further elucidation and attention.

Thank you for your positive comments

Major comments:

  1. Please furnish the immunological records of the animals used in this study, to the extent of the author's knowledge. While lines 94-96 detail the immune records of a guinea pig, there is no mention of other animals, particularly domesticated ones such as cats and dogs.

Answer: Cat and dog samples were remnants from clinical analyses of two veterinary clinics in Hermosillo, Sonora, unfortunately, clinical information of these samples is unavailable.

  1. It has come to attention that two distinct figures are both labeled as "Figure 2", one on page 7 and the other on page 10. This has led to confusion among the readers. Please rectify this oversight.

Answer: This mistake has been corrected accordingly.

  1. Pertaining to "Figure 2" on page 7, the concentration of the capture antigen concentration group at 0.5 ug/mL shows the bar at 0.4 (blue bar) being higher than at 0.8 (yellow bar). A similar observation is made for the 1.0 ug/mL and 2.0 ug/mL groups. Please provide an explanation and delve into a discussion on this phenomenon.

Answer. The explanation was included in the discussion section as suggested.  Lines 427-433

  1. In Table 1, the diagnostic sensitivity of the N protein stands at a mere 60%, with a p-value of 0.2226. It would be beneficial if the authors could discuss this in the subsequent section, perhaps shedding light on the reasons for the N protein's performance. Similarly, Table 2 indicates that the indirect ELISA with the S1 protein exhibits low diagnostic sensitivity. An explanation for this observation would be appreciated.

Answer. The explanation was included in the discussion section as suggested.  Lines 472-482

  1. Between lines 357 and 359, it is mentioned that 3 out of the 45 dog samples tested positive using the sandwich ELISA method. It would be informative to know whether these 3 samples had previously tested positive through other techniques, whether the respective dogs exhibited any positive symptoms, or if there's a possibility that these were false positives.

Answer: Because dog samples were remnants from clinical analyses, we lacked information regarding previous positive symptoms, and these samples could not be further characterized. According to our results and the performance of the double antigen sandwich ELISA method, we can stand that the samples have no false positives.

Minor Comments:

  1. "Microneutralization test" is abbreviated as both "MNT" and "MN test". It's recommended to consistently use one abbreviation throughout.

Answer: We have homogenized the use of “MNT” abbreviations in the manuscript

  1. Line 185: Insert a space before starting a new paragraph.

Answer: We have modified this aspect in the manuscript

  1. Line 457: "ELISAS" should be corrected to "ELISAs".

Answer: We have modified this aspect in the manuscript

Moderate editing of English language required.

Round 2

Reviewer 3 Report (New Reviewer)

Comments and Suggestions for Authors

The authors have significantly improved the manuscript. After minor revision, the manuscript is ready for acceptance. I report some sentences that should be changed.

Line 17-19: Rephrase it

Line 24: "this assay detects" in "this assay is able to detect" or similar

Line 31: There is an extra "ideal"?

Line 39: "This multispecies analysis increased the power of the test". Please remove it.

Line 41: "(n=)" ?

Line 110: Please, avoid statement as "perfmored by our group" or "our works". 

Line 132-135: Ripetition?

Line 502: "The use of the N protein to validate an indirect ELISA seems difficult," please change it.

Line 519: "This is similar to our findings" as Line 110 

Comments on the Quality of English Language

Moderate English proofreading should be done to make the manuscript more easily comprehensible. Some periods are long and complicated. Others short and obvious.

Reviewer 4 Report (New Reviewer)

Comments and Suggestions for Authors

N/A

Comments on the Quality of English Language

N/A

This manuscript is a resubmission of an earlier submission. The following is a list of the peer review reports and author responses from that submission.

Round 1

Reviewer 1 Report

Comments and Suggestions for Authors

Ref: animals-2537451
Title:
Development of a multispecies double-antigen sandwich 2 ELISA using N and RBD proteins to detect antibodies against 3 SARS-CoV-2

The study aimed to develop a double-antigen sandwich ELISA using two SARS-CoV-2 proteins, N and RBD. The authors compared the performance of the developed with an indirect ELISA and surrogate virus neutralization test using 31 cat serum samples. Sensitivity, specificity, and AUC values based on ROC curves were obtained using positive and negative samples from 37 cats, humans, and guinea pigs to adjust the cutoff and increase the power of the test. When samples 38 of tigers and rats were evaluated, good agreement was found with the microneutralization test. According to the results, using two SARS-CoV-2 proteins in the double-antigen sandwich ELISA increases its performance.

1)      Defining a research problem is crucial, but the research problem of this study needs to be better defined. According to the authors, the plaque reduction neutralization test (PRNT), which is the gold standard for detecting the presence of neutralizing antibodies against SARS-CoV-2, and the microneutralization test (MNT) are time-consuming, requiring level 3 biosafety laboratory (BSL3) conditions and well-trained personnel. Therefore, as a justification, the authors developed this double-antigen sandwich ELISA. However, as was emphasized by the authors themselves, this technique has already been developed and is even commercially available and utilized in at least five publications. To justify developing this technique, the authors should emphasize the difference from the available double-antigen sandwich ELISAs, particularly the fact that SARS-CoV-2 RBD-based assays feature higher sensitivity and specificity compared to N protein assays and the possibility that the IDVet Screen ELISA positives were most likely due to exposure to other animal coronaviruses. Please note that the presence of antibodies against the spike protein in the absence of antibodies against nucleocapsid proteins is reported in people.

2)      The methodology needs a critical step. The study aimed to develop a double-antigen sandwich ELISA. For such a purpose, the authors should let the readers know how they reach the optimum condition described in the methodology. Optimizing an assay is fundamental and critical to achieving a quality assay. In addition to sensitivity and specificity, the evaluation should include accuracy, repeatability, and reproducibility.

3)      More samples than those used (control positive, negative, and field samples) are required for adequate validation. The investigators must know the minimum sample size needed for their research to estimate a test's diagnostic accuracy and achieve the requisite statistical power. There are various methods for estimating sample sizes. Collecting sufficient samples from SARS-CoV-2 negative, infected, or immunized animals is essential to assess the ELISA's performance.

4)      Details of the materials used to develop the test are unavailable. The authors used SARS-CoV-2 proteins, N and RBD, without proper details on their origin or characteristics that directly affect the tests' performance characteristics. I accessed reference 16 (Melgoza-González et al., 2022), assuming I would find the needed information. Unfortunately, the reference detailed the preparation of SARS-CoV-2 S1 protein only. 

5)      The methodology of this study needs to be revised. The methodology must be provided to arrive at the sensitivity, specificity, and AUC value. Reporting of data and methodology must be sufficiently detailed and transparent to enable the reproduction of the results.    Standard methods estimate the sensitivity and specificity of a diagnostic test by comparing the results of a new test to the results of a ‘gold standard’ reference test. Did the authors estimate the sensitivity and specificity of the developed test compared to the PRNT?  Is the PRNT for SARS-CoV-2 a ‘gold-standard’ or perfect test, with 100% sensitivity and 100% specificity? The study will significantly benefit if the developed test is compared with the commercially available double-antigen sandwich ELISA (ID VET).

Comments on the Quality of English Language

No comment

Reviewer 2 Report

Comments and Suggestions for Authors

Diagnostic test for the detection of viral infection in animal is important for the risk of interspecies transmission and the emergence of new viral variants. Therefore, sensitive diagnostic tests are needed to monitor the emergence of infection. In this study the authors developed and evaluated the performance of a double-antigen sandwich ELISA using two SARS-CoV-2 proteins, N and RBD. The results are interesting, but the main limit of the study is the small size of positive samples. Although this limit is difficult to overcome in veterinary field, the authors should discuss this aspect.

The availability of serological tests can help to define the epidemiology of an infection and identify individuals previously infected, but this approach cannot be used to identify animals acutely ill. The authors should better underline the potential use of their assay.

In microneutralization test 5 cat samples resulted positive: 3 had a titer 1/10, 1 had a titer 1/20 and the last 1/40. Was there a relationship between MN titer and absorbance value?

Paragraph 3.2: The sentence “As mentioned before, 31 189 cat samples were tested, resulting in positive (n=5) and negative (n=26) results (MN titer: positive 1:10; negative <1:10)” should be omitted

Reviewer 3 Report

Comments and Suggestions for Authors

The terms sensitivity and specificity as used in this paper need to be further identified as diagnostic sensitivity and diagnostic specificity to differentiate them from analytical sensitivity and analytical specificity.

The cutoffs determined by ROC curves can be adjusted so that all true positive samples test positive. However, increasing the sensitivity by this means will reduce the specificity. This adjustment may be done depending on which results are more important, false positives or false negatives.

 Sera from different species may have different cut-offs in the tests, so the use of only cat and rat sera to determine true positives and negatives in the microneutralization test may not necessarily be applied to other species.

To determine the true status sera tested in this trial all sera should have been tested by the microneutralization test as well as in the various tests described in this paper.